# Reframing the contribution of pelagic *Sargassum* epiphytic N₂ fixation

**Claire Johnson**[1]*, **Lindsay L. Dubbs**[1,2,3], **Michael Piehler**[2,4]

**1** Environment, Ecology and Energy Program, University of North Carolina at Chapel Hill, Chapel Hill, North Carolina, United States of America, **2** Institute for the Environment, University of North Carolina at Chapel Hill, Chapel Hill, North Carolina, United States of America, **3** Coastal Studies Institute, East Carolina University, Wanchese, North Carolina, United States of America, **4** Earth, Marine and Environmental Sciences Department, University of North Carolina at Chapel Hill, Chapel Hill, North Carolina, United States of America

* clairejo@unc.edu

**Data Availability Statement:** All relevant data are within the manuscript and its Supporting Information files.

**Funding:** North Carolina Renewable Ocean Energy Program to LD and MP (no grant number); https://

## Abstract

Though nitrogen fixation by epiphytic diazotrophs on pelagic *Sargassum* has been recognized for decades, it has been assumed to contribute insignificantly to the overall marine nitrogen budget. This six-year study reframes this concept through long-term measurements of *Sargassum* community nitrogen fixation rates, and by extrapolating mass-specific rates to a theoretical square meter portion of *Sargassum* mat allowing for comparison of these rates to those of other marine and coastal diazotrophs. On 24 occasions from 2015 to 2021, rates of nitrogen fixation were measured using whole fronds of *Sargassum* collected from the western edge of the Gulf Stream off Cape Hatteras, North Carolina. Across all dates, mass-specific rates ranged from 0 to 37.77 μmol N g⁻¹ h⁻¹ with a mean of 4.156 μmol N g⁻¹ h⁻¹. Extrapolating using a mat-specific density of *Sargassum*, these rates scale to a range of 0 to 30,916 μmol N m⁻² d⁻¹ and a mean of 3,697 μmol N m⁻² d⁻¹. Quantifying this community's rates of nitrogen fixation over several years captured the sometimes-extreme variability in rates, characteristic of marine diazotrophs, which has not been reported in the literature to date. When these measurements are considered alongside estimates of the density of pelagic *Sargassum*, rates of nitrogen fixation by *Sargassum*'s epiphytic diazotrophs rival that of their coastal macrophyte and planktonic counterparts. Given *Sargassum*'s wide and expanding geographic range, the results of this study suggest this community may contribute reactive nitrogen on a meaningful, basin-wide scale, which merits further study.

## Introduction

In the open ocean, nitrogen fixation is one of the largest sources of biologically available nitrogen [1]. This newly fixed nitrogen is critical to primary production in the euphotic zone, where it is often limiting [2]. Over the last several decades, knowledge of the magnitude of marine nitrogen fixation [3, 4] as well as the diversity [5] and ecology (reviewed by [6]) of marine nitrogen fixers has greatly expanded. Today, we know marine diazotrophs to be

www.coastalstudiesinstitute.org/ncroep/; The funders had no role in study design, data collection and analysis, decision to publish, or preparation of the manuscript. National Science Foundation Graduate Research Fellowship to CJ (DGE-1650116); https://www.nsfgrfp.org; The funders had no role in study design, data collection and analysis, decision to publish, or preparation of the manuscript.

**Competing interests:** The authors have declared that no competing interests exist.

genetically and physiologically diverse, with cyanobacterial and non-cyanobacterial representatives found in various morphologies across virtually all marine habitats from shallow coastal areas to the open ocean and deep sea [7].

While planktonic representatives may be the archetype of marine diazotrophs, nitrogen fixers also exist epiphytically, commonly colonizing the surfaces of macroalgae, seagrasses, and mangroves [8–10]. Such macrophyte-associated diazotrophs are an important source of nitrogen to the local community and can contribute significantly to host nutrition (e.g., [11–13]). *Sargassum*, a cosmopolitan genus of brown macroalgae, is found in marine waters throughout the world primarily inhabiting benthic coastal environments. Unique among aquatic macroalgae, however, are two species of *Sargassum*, *S. fluitans* and *S. natans*, that exist in a pelagic, unattached form. Typically found in the Caribbean Sea, Gulf of Mexico, Gulf Stream, and Sargasso Sea [14], pelagic *Sargassum* aggregates in large mats and windrows, sometimes kilometers long, but also exists in smaller, more sparsely distributed patches and clumps [15]. Pelagic *Sargassum* is unique in the open ocean because it provides structure in an environment otherwise devoid of such complexity. As such, it is a locus of biologic and chemical activity serving as a habitat and foraging site to seabirds [16], hatchling and juvenile sea turtles [17], and a diverse assemblage of fish and invertebrates [18, 19], some of which are endemic. Perhaps less well known is the diverse epibiotic, and particularly microbial community *Sargassum* supports. Common amongst its epibiota are nitrogen-fixing cyanobacteria, which have been known contributors of reactive nitrogen to this system for over 50 years [20]. In the context of what was known about marine nitrogen fixers, namely *Trichodesmium*, and the few studies available at the time, nitrogen inputs by pelagic *Sargassum*'s epiphytic diazotrophs were considered insignificant to the overall marine nitrogen budget [21].

In the last decade, however, pelagic *Sargassum* has been observed in large quantities outside its typical range in the North Atlantic, forming what is known as the Great Atlantic Sargassum Belt (GASB) almost every summer since 2011 [22]. Stretching from the west coast of Africa to the Caribbean and Gulf of Mexico [22], these annual *Sargassum* blooms are believed to be related to a strong wind event in the winter of 2009–2010 that forced *Sargassum* out of the Sargasso Sea and into the tropical Atlantic where favorable conditions allowed it to proliferate [23]. This single incident has retained *Sargassum* within the tropical Atlantic allowing blooms to form almost every summer since and is the likely cause of the large, disruptive inundations of the macroalgae on Caribbean and Southeastern US beaches [23]. Given the status of nitrogen in the oligotrophic ocean, epiphytic diazotrophs almost certainly play a critical role in the productivity of the *Sargassum* community within the GASB [24] and beyond.

Considering the sheer volume of pelagic *Sargassum*, its epiphytic community's ability to fix nitrogen, and the methodological constraints of previous studies, we believe nitrogen fixation by this community is underestimated and may contribute reactive nitrogen on a scale that is relevant to the nitrogen budget of the Atlantic. The goal of this study was to contextualize reactive nitrogen inputs from nitrogen fixation to the *Sargassum* community. To achieve our goal, we collected a long-term dataset of nitrogen fixation rates (mass-specific) by pelagic *Sargassum*'s epiphytic diazotroph community using the acetylene reduction technique and extrapolated those rates to a theoretical square meter *Sargassum* mat. This type of mat-specific extrapolation is essential to accurately compare nitrogen fixation rates among the wide range of studies that span benthic coastal to planktonic marine environments and facilitates inclusion of nitrogen fixation by pelagic *Sargassum* epiphytes into biogeochemical models. The acetylene reduction method was chosen because it provides the most direct comparison to other pelagic *Sargassum* community studies, all of which have used this technique, and because of its ubiquity in benthic macrophyte studies. Additionally, as acetylene reduction is a proxy measure of gross nitrogen fixation, it quantifies reactive nitrogen inputs to the entire

community rather than the alternative $^{15}$N$_2$ tracer technique, which measures net nitrogen fixation. Furthermore, $^{15}$N$_2$ gas stocks have been found to contain considerable amounts of biologically available $^{15}$N nitrogen species which may artificially inflate rates of nitrogen fixation [25].

Here, we present the results from a six-year (2015–2021) study of nitrogen fixation rates by pelagic *Sargassum*'s epiphytic diazotrophs in the Gulf Stream off Cape Hatteras, North Carolina It reveals this community's potentially significant role in regional marine nitrogen cycling.

## Materials and methods

### Sample collection

*Sargassum* is reliably found in varying quantities and mat morphologies throughout the year along the western edge of the Gulf Stream off Cape Hatteras, North Carolina. *Sargassum* in this area was collected seasonally on 24 occasions over a period of six years from June 2015 to July 2021 (Fig 1 and Table 1). A Gulf Stream location was identified by water temperature and salinity measurements (YSI 6600 V2). Loose macrofauna incidentally collected were carefully removed before *Sargassum* was placed in insulated 5-gallon buckets filled with unfiltered seawater. During transport back to the laboratory (typically 7–8 hours), buckets were left partially open for oxygen exchange and exposure to sunlight.

Upon return to the laboratory, *Sargassum* was placed in 102 L cylindrical tubs filled with 95 L of local, unfiltered seawater within a temperature-controlled environmental chamber set to replicate *in situ* water temperatures (23-28˚C). A diurnal light cycle was maintained by connecting two 400-watt high pressure sodium lamps (approximately 240 μmol photons s$^{-1}$ m$^{-2}$,

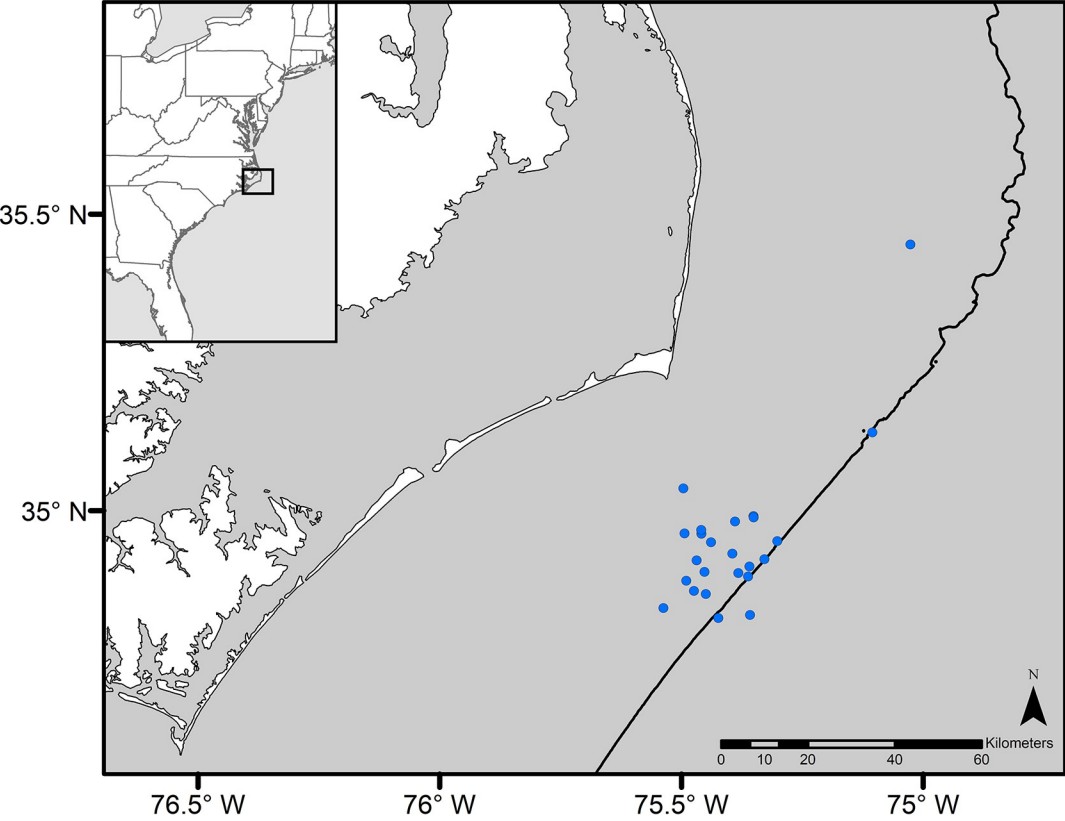

**Fig 1. Collection locations of pelagic *Sargassum* used in AR assays.** Solid black line represents the 250-meter isobath [26].

**Table 1. *Sargassum* collection dates, locations, mean mass-specific rates of nitrogen fixation, and number of replicate assays (n).**

| Date | Location | | Mean $N_2$ fixation ± SE ($\mu$mol N $g^{-1}$ $h^{-1}$) | n |
|---|---|---|---|---|
| | Latitude | Longitude | | |
| 16 June 2015 | 35.04 | -75.50 | trace | 2 |
| 09 October 2015 | 34.86 | -75.47 | 1.016 ± 0.0221 | 2 |
| 17 November 2015 | 34.92 | -75.33 | trace | 2 |
| 16 May 2016 | 34.98 | -75.39 | 1.897 ± 1.594 | 2 |
| 25 May 2016 | 35.13 | -75.11 | 17.12 ± 7.377 | 2 |
| 15 June 2016 | 34.96 | -75.46 | 1.958 ± 1.339 | 2 |
| 14 September 2016 | 34.89 | -75.36 | trace | 2 |
| 16 November 2016 | 34.86 | -75.45 | 0 | 2 |
| 09 March 2017 | 34.89 | -75.38 | 0.6078 ± 0.3376 | 2 |
| 10 May 2017 | 34.90 | -75.45 | 3.985 ± 2.748 | 3 |
| 17 May 2017 | 34.82 | -75.36 | 2.024 ± 0.6073 | 3 |
| 03 August 2017 | 34.95 | -75.30 | 0.7435 ± 0.3175 | 2 |
| 01 November 2017 | 34.93 | -75.39 | 0 | 3 |
| 22 May 2018 | 34.92 | -75.47 | 3.170 ± 0.3743 | 3 |
| 7 August 2018 | 34.96 | -75.49 | 1.348 ± 0.3549 | 3 |
| 14 March 2019 | 34.99 | -75.35 | 0.7680 ± 0.2285 | 3 |
| 04 June 2019 | 35.45 | -75.03 | 6.735 ± 0.7648 | 3 |
| 24 February 2020 | 34.82 | -75.42 | trace | 3 |
| 26 May 2020 | 34.99 | -75.35 | 10.74 ± 3.864 | 3 |
| 28 September 2020 | 34.84 | -75.54 | 12.96 ± 2.797 | 3 |
| 10 March 2021 | 34.91 | -75.36 | 5.832 ± 1.385 | 3 |
| 27 April 2021 | 34.88 | -75.49 | 5.472 ± 1.496 | 3 |
| 07 June 2021 | 34.95 | -75.44 | 32.34 ± 5.435 | 2 |
| 23 July 2021 | 34.97 | -75.46 | 1.346 ± 0.6089 | 3 |

Trace amounts of nitrogen fixation occurred when all replicates had rates less than 0.5 $\mu$mol N $g^{-1}$ $h^{-1}$ and/or at least one replicate had a rate of 0 $\mu$mol N $g^{-1}$ $h^{-1}$. See S1 for all replicate data.

PAR) to timers. Water flow within the tubs was simulated using circulation pumps (Sicce, Voyager 2, maximum flow rate approximately 3 $m^3$ $hr^{-1}$). To reduce the chance of experimental artifacts from transport-induced stress, *Sargassum* was left to adjust to these conditions overnight (12–14 hours).

## Acetylene reduction assays

For the purpose of our goal, understanding reactive nitrogen inputs to the pelagic *Sargassum* community, we measured rates of nitrogen fixation using the acetylene ($C_2H_2$) reduction (AR) method [27], a measure of gross nitrogen fixation [28]. *Sargassum* fronds (15–25 g wet weight, 2–4 g dry weight) were placed in 500 mL glass media bottles filled with approximately 405 mL of seawater in duplicate or triplicate. Incubations were begun by replacing 12% of bottle headspace with $C_2H_2$ (generated from calcium carbide in the laboratory within one hour of the start of each experiment) and removing a headspace subsample. Bottles were equilibrated by several gentle inversions and incubated under high pressure sodium lamps (approximately 240 $\mu$mol photons $s^{-1}$ $m^{-2}$, PAR) in water baths at *in situ* temperatures. A final headspace subsample was taken after 3 or 4 hours. *Sargassum* from each incubation bottle was weighed after drying at 60°C for at least 48 hours.

Gas samples were analyzed for ethylene (C$_2$H$_4$) concentration by flame ionization detection gas chromatography (Shimadzu GC-2014). After correcting for background C$_2$H$_4$ in the added C$_2$H$_2$ and solubility of C$_2$H$_4$ in the liquid phase [29], C$_2$H$_2$ reduction rates were calculated by linear regression of C$_2$H$_4$ concentration over the duration of incubation. Rates of nitrogen fixation were calculated using a 3:1 C$_2$H$_4$:N$_2$ ratio and multiplied by 2 to express rates as fixed nitrogen. Rates are reported on a per gram dry weight *Sargassum* basis [27].

On each sampling occasion, controls for both ethylene production by *Sargassum* and acetylene reduction by planktonic diazotrophs were made to ensure ethylene concentrations were only attributable to *Sargassum*'s epiphytic diazotrophs. Ethylene production never occurred in controls and was only measured in experimental treatments.

## Data analysis

Anomalously high rates were identified prior to analysis using Chauvenet's criterion and removed. Not unlike most ecological data, our dataset is zero-inflated and violates this method's assumption of normality. We are, however, only concerned with right tail (high rate) outliers as the high frequency of zero observations are true zeros, or instances of diazotroph absence or inactivity. For this reason and exclusively for the purpose of this analysis, only non-zero observations were considered [30].

Non-zero data was log-transformed and values with a probability greater than 1/(4n) were removed based on mean and standard deviation. Although no values exceeded this critical value, one value, the highest observed rate, came very close. As this rate was over three times greater than the next highest rate, we conservatively removed it.

After outlier removal, the original (non-transformed) data, including zero observations, was grouped by cruise date and summary statistics (minimum and maximum observed rates, mean, and standard error) were calculated (Table 1).

## Rate extrapolation

Rates of nitrogen fixation by marine planktonic and coastal epiphytic diazotrophs are typically expressed on a daily, areal basis (flux). To contextualize our results, we extrapolated our hourly mass-specific rates to daily mat-specific (m$^2$) rates.

Like previous studies, [12, 20, 31], we observed little to no nitrogen fixation in the dark (unpublished), which suggests the community is comprised, at least in part, of cyanobacteria. For this reason, we conservatively assumed nitrogen fixation occurs only during daylight hours (approx. 12 h$^{-1}$ d$^{-1}$). Rates of fixation, however, are not constant throughout the photoperiod, so daily rates were integrated using a bell-shaped curve [32] to account for the "ramping up" and "ramping down" of nitrogen fixation that has been observed in light-dependent diazotrophy [33–35].

Previous studies [20, 31] extrapolated mass-specific rates of nitrogen fixation using regionally averaged densities of *Sargassum* published by Parr [36], whose goal was to estimate the standing crop (total biomass) of pelagic *Sargassum* in different regions of the Atlantic. By towing a net of known size across the length of a transect and weighing the *Sargassum* that was collected, Parr was able to calculate an average biomass per unit area. These density estimates were made by sampling areas both present and absent of *Sargassum*, essentially "evening out" *Sargassum* biomass over an entire region. While density measurements made using the net towing method are useful in determining *total* biomass of *Sargassum* within a region, it is not useful in calculating nitrogen inputs per area mat of *Sargassum*, which was our goal.

This distinction is especially important when comparing areal rates of nitrogen fixation by the *Sargassum* community with that of other marine and coastal diazotrophs as areal rates are

calculated using time and location-specific densities measured *simultaneously* with rates. In the case of marine planktonic diazotrophs, location-specific rates are depth integrated either by making discrete measurements at specific depths (e.g., [2]) or by measuring colony or tri-chome-specific rates and extrapolating using location and depth-specific abundance (e.g., [37]). For epiphytic diazotrophs on coastal benthic macroalgae, local macroalgal density is either directly measured (e.g., [38]) or inherently taken into account by incubating within vessels of known size *in situ* (e.g., [39]). To accurately compare these areal rates of nitrogen fixation to that of pelagic *Sargassum*, density estimates for *Sargassum* need to be made in a similar way. For this reason, we used a mat-specific density to extrapolate our mass-specific rates.

Wang et al. [40] reported an average mat-specific density of 3,340 g wet weight *Sargassum* m$^{-2}$, however evidence suggests larger quantities of *Sargassum* exist in the Gulf of Mexico, with biomass decreasing as it is advected into the Loop Current and is carried north by the Gulf Stream [14, 40]. Assuming this trend continues northward to Cape Hatteras, we conservatively used the minimum mat-specific *Sargassum* density reported by Wang et al. (1,260 g wet weight m$^{-2}$) [40] as a proxy for *Sargassum* density in our study area. A mat-specific dry weight density of 186.6 g m$^{-2}$ was calculated by linear regression of paired wet and dry *Sargassum* weights resulting in a conversion factor of 6.75.

## Results

Over the course of six years and 24 cruises, mass-specific hourly rates of nitrogen fixation ranged from 0 to 37.77 μmol N g$^{-1}$ (dry weight *Sargassum*) h$^{-1}$ (07 June 2021) with a mean (± standard error) of 4.156 ± 0.9243 μmol N g$^{-1}$ h$^{-1}$ (n = 61). While approximately one third of all observations exceed this mean rate, we observed little (maximum rate ≤ 0.4 μmol N g$^{-1}$ h$^{-1}$) to no nitrogen fixation on six dates (Table 1).

Integrating hourly mass-specific rates over a 12-hour photoperiod using a bell-shaped curve and multiplying by dry weight mat-specific density result in daily areal rates averaging 3,697 μmol N m$^{-2}$ d$^{-1}$ with a range of 0 to 30,916 μmol N m$^{-2}$ d$^{-1}$.

Mass-specific rates of nitrogen fixation are quite variable both among cruises and among replicates. Three cruises made over the course of just one month in 2016 (16 May 2016 to 15 June 2016) demonstrate how strikingly different rates of nitrogen fixation can be on short time scales (Fig 2A). This same kind of extreme variability was observed again in June and July 2021. Variation among samples from a single cruise can also be quite large and, in some cases, can vary by an order of magnitude. In general, standard error among replicates is greater when the range of observations crosses or falls entirely above the mean rate than when it falls entirely below (Fig 2B). This implies greater variability in instances of relatively high rates relative to instance of low rates.

## Discussion

### Comparison to other pelagic *Sargassum* studies

Since Carpenter's discovery of diazotrophs living epiphytically on pelagic *Sargassum* in 1972 [20], only four studies have been published which directly measured rates of nitrogen fixation by this community (Table 2). We were unable to compare nitrogen fixation rates from this study with those of the other primary studies because we did not have access to raw datasets, and the published minimums, maximums, and averages provided too small a dataset. Instead, we contextualize observed rates of nitrogen fixation with the methods used to measure them. Furthermore, we limit our discussion to cyanobacterial diazotrophs as this is the only group identified and implicated by all previous studies (Table 2).

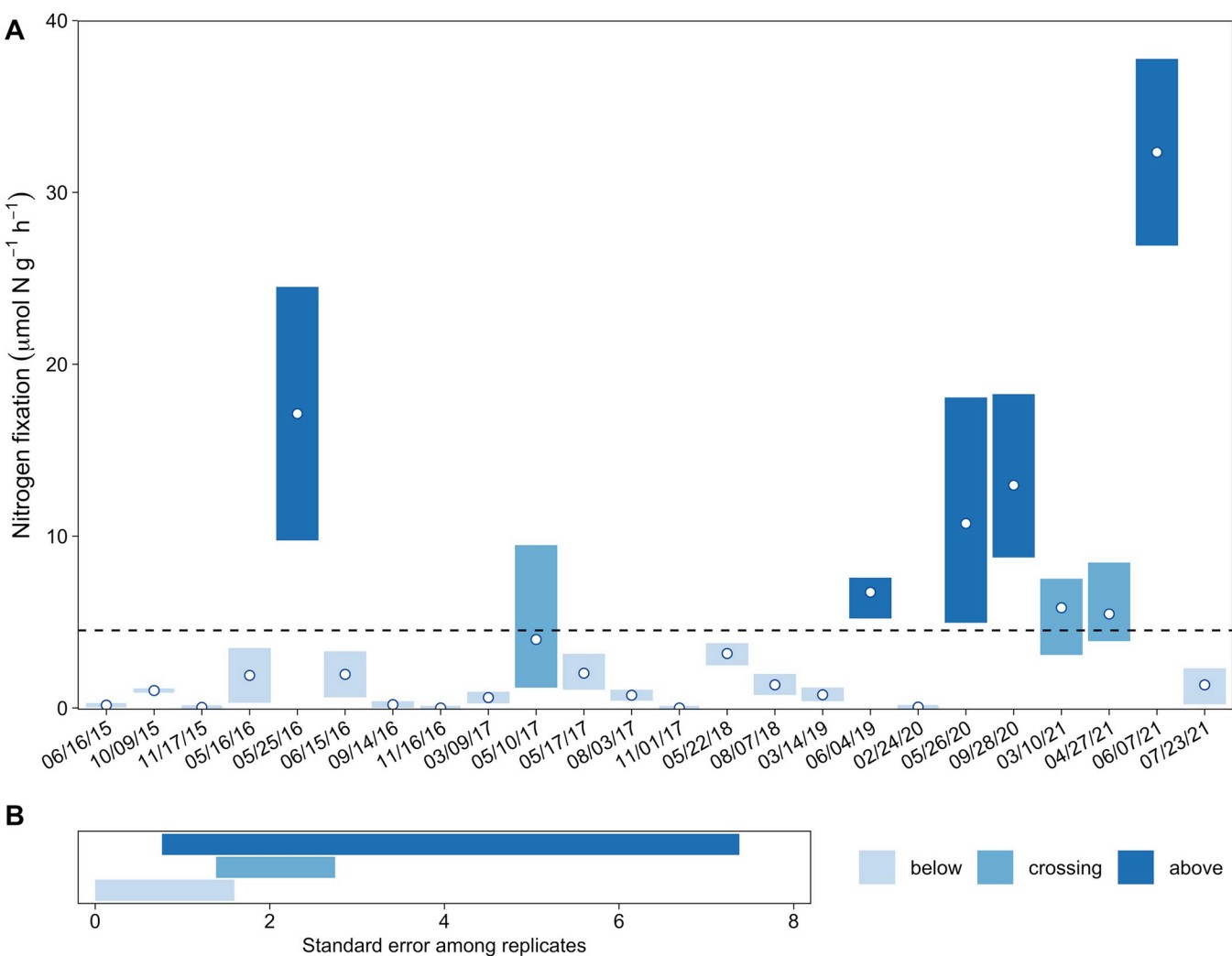

**Fig 2.** Range (colored bars) and mean (open circles) nitrogen fixation rates by cruise date (**A**) and standard error range among replicates by category (**B**). Bar color in panel A indicates whether a cruise date's range of rates falls entirely below, crosses, or entirely above the overall mean rate (dashed line). Panel B illustrates the range of standard error among replicates from a single cruise date according to these same mean rate categories.

Of the four published studies detailing rates of nitrogen fixation by pelagic *Sargassum*'s epiphytic diazotrophs, none assayed whole fronds of *Sargassum* as we have done here. By doing this, we have minimized handling *Sargassum*, which is important as any kind of manipulation (holding, shaking, cutting, etc.) can preferentially remove loosely attached epiphytes and disrupt the integrity of the community's microstructures [42]. Reduced microzones within biofilms such as these are integral to maintaining nitrogen fixation [43, 44] as the metalloproteins of nitrogenase, the enzyme that catalyzes N$_2$ reduction, are degraded by molecular oxygen [45]. Damage of this nature is especially significant if non-heterocystous cyanobacteria are present as they have no specialized physical structures to separate nitrogenase from oxygen. Additionally, whole plant assays are considered a more accurate representation of nitrogen fixation activity [43].

Although both Hanson [31] and Phlips et al. [12] assayed whole pieces of *Sargassum*, incubations were done in small ($\leq$ 50 mL) flasks, which required excising the plant into small fragments (adding up to no more than 3 g wet weight). It is possible the prolonged handling and cutting required by this method not only damaged the finely structured microzones and

**Table 2. Rates of nitrogen fixation by epiphytic diazotrophs on pelagic *Sargassum* in reverse chronological order.** Rates are highest reported values (unless otherwise noted) under any condition.

| Location | *Sargassum* community element assayed | Diazotroph(s) identified and/or suggested | N$_2$ fixation | | Method | Comment | Source |
|---|---|---|---|---|---|---|---|
| | | | µmol N g$^{-1}$ h$^{-1}$ min–max (avg.) | µmol N m$^{-2}$ d$^{-1}$ min–max (avg.) | | | |
| Gulf Stream | *Sargassum* with intact epiphytes | not identified | 0–37.77 (4.156) | 0–30,916 (3,697) | AR, 3:1 | | this study |
| Florida Straits | removed epiphytes | *Oscillatoria* isolates | 0–47.8 (5.2) | – | AR, 3:1[a] | among all strains, anaerobic conditions resulted in highest rates | [41] |
| Gulf Stream | *Sargassum* with intact epiphytes | filamentous (heterocystous and non-heterocystous) and coccoid cyanobacteria | 0.05–4.7 | – | AR, 3:1 | | [12] |
| summary derived from [20, 31] | | | (0.088)[b] | – | AR, 3:1[c] | | [21] |
| Sargasso Sea and Gulf Stream | *Sargassum* with intact epiphytes | cyanobacteria | 0.117–3.41 (1.1) | (3.2)[d, e] | AR, 3:1 | rates up to 51.88 µmol N g$^{-1}$ h$^{-1}$ were observed on "unhealthy plants" | [31] |
| | removed epiphytes | *Dichothrix fucicola* | – | 0–0.771[e] | AR, 3:1 | | [20] |

[a] N$_2$:C$_2$H$_4$ ratio not reported, assuming 3:1

[b] originally reported as a daily rate, converted to hourly rate (12 h d$^{-1}$)

[c] rate derived from [20, 31], both of which used the AR technique and a 3:1 N$_2$:C$_2$H$_4$ ratio

[d] originally reported as an hourly rate, converted to daily rate (12 h d$^{-1}$)

[e] calculated using Parr's [36] *Sargassum* biomass estimate

macroalgal cells but may have also differentially removed epiphytic cells. Despite the limitations of this method, Hanson [31] and Phlips et al. [12], measured rates as high as 3.41 and 4.7 µmol N g$^{-1}$ h$^{-1}$, respectively (Table 2). The similarity in rates measured in these two studies may reflect their almost identical methodology. In contrast, we assessed whole, intact diazotroph community rates of nitrogen fixation by carefully handling and assaying whole fronds of *Sargassum*, which is likely why we observed significantly higher rates.

Other studies measuring rates of nitrogen fixation isolated specific diazotrophs found on *Sargassum*. In an effort to demonstrate the nitrogen fixing capability of *Dichothrix fucicola*, a heterocyst-bearing cyanobacteria discovered living epiphytically on pelagic *Sargassum*, Carpenter [20] selectively removed and then assayed the diazotroph in filtered seawater suspensions. Although removal from *Sargassum* should not theoretically affect rates of nitrogen fixation by *D. fucicola*, as it is heterocystous, Carpenter [20] measured a maximum rate of 0.771 µmol N m$^{-2}$ d$^{-1}$ (based on Parr's 1939 wet weight standing crop of *Sargassum* and a wet to dry weight conversion factor of 5.9, quite similar to ours; Table 2). This is the *D. fucicola*-specific rate and is, therefore, not representative of the wider, much more diverse assemblage of epiphytic diazotrophs known to live on pelagic *Sargassum* [12].

Like Carpenter [20], Phlips and Zeman [41] also removed epiphytes from pelagic *Sargassum*, though they were isolated and then cultured on low-nitrogen media. After two weeks of growth, rates of nitrogen fixation were measured under a variety of physical and chemical conditions. Of the five strains that actively fixed nitrogen, later identified as *Oscillatoria* spp., higher rates were measured among anaerobic rather than aerobic treatments. This is not surprising given *Oscillatoria* is a non-heterocystous cyanobacteria. These findings highlight the importance of maintaining *Sargassum*'s delicate biofilm as it likely provides the low-oxygen conditions required by certain diazotrophs to protect nitrogenase. In anaerobic light conditions, Phlips and Zeman [41] measured fixation rates by these isolates up to 47.8 µmol N g$^{-1}$ h$^{-1}$, similar in

magnitude to rates we observed (Table 2). As with Carpenter's [20] results, these rates are *Oscillatoria*-specific and likely underestimate whole community rates of nitrogen fixation.

Consistent with what we have observed among epiphytes on pelagic *Sargassum*, rates of nitrogen fixation by cyanobacteria are generally quite variable and wide ranging [30]. Particularly characteristic of macroalgae, this variability is usually attributed to the patchy nature of epiphyte colonization throughout the plant [12, 46, 47] but may also be due to other environmental or diazotroph-specific variables [48]. Unlike other macroalgae, pelagic *Sargassum* exists in the low-nutrient environment of the open ocean which is subject to unpredictable and irregular pulses of limiting nutrients [49–51], further amplifying the inconsistent nature of nitrogen fixation within this community. Long-term studies are best suited to capture this type of complex variability, though the cost and logistics of marine research make this a challenge.

Phlips et al. [12] published the only other long-term pelagic *Sargassum* nitrogen fixation study, which took place over the course of two years. Interestingly, the significant month-to-month variability in rates observed among pelagic *Sargassum* samples was not evident in its benthic (*S. filipendula*) counterpart [12], further underpinning the importance of achieving fine temporal resolution in oligotrophic systems.

Common amongst macroalgal, primarily benthic systems, mass-specific rates of nitrogen fixation are useful in making cross-system comparisons. However, to contextualize these fine-scale inputs within the larger system, extrapolations are required (i.e., quantity of nitrogen on a yearly or areal basis). In an effort to understand and quantify sources of nitrogen fixation in the marine environment, Capone and Carpenter [21] considered epiphytes on pelagic *Sargassum* to be a relatively unimportant source, especially when compared to the much more widely and frequently studied cyanobacteria, *Trichodesmium*. The estimate given for pelagic *Sargassum* (0.088 μmol N g$^{-1}$ h$^{-1}$ or 1.1 x 10$^6$ mol N year$^{-1}$), however, was based on rates measured by Carpenter [20] and Hanson [31], the only two studies available at the time. In addition to the methodological constraints previously discussed, both studies were temporally limited. While these two initial studies made significant progress in advancing knowledge about the identity and environments in which marine diazotrophs are found, both were effectively snapshots in time with sampling taking place over the course of just a few days. Because rates of nitrogen fixation vary so widely, a time series of data as short as these do not capture the full scope of variability within the system and demonstrate the importance of using long-term datasets to inform larger, basin-scale extrapolations and model inputs.

Although areal rates of nitrogen fixation have been made for pelagic *Sargassum* [20, 31], these estimates were calculated using the aforementioned net towing method to measure *Sargassum* density [36], which, by definition, underestimates mat-specific density. Location-specifc densities, in contrast to regionally averaged densities, are required to accurately compare areal rates of nitrogen fixation by the pelagic *Sargassum* community with those of other marine planktonic and coastal macrophyte studies which use time and location-specific abundance and/or density measurements. Calculating areal rates using a mat-specific density allows us to put into perspective the magnitude of reactive nitrogen inputs by the pelagic *Sargassum* community withinin the extensive framework of literature which reports areal rates of nitrogen fixation. To our knowledge, this is the first report of such an estimate for pelagic *Sargassum*. For this reason, it is not surprising that our areally extrapolated values are orders of magnitude higher than both Carpenter's [20] and Hanson's [31] (Table 2).

## Comparison to epiphytic diazotrophs on macrophytes

Coastal seagrass meadows are generally considered hotspots of nitrogen fixation, with a substantial proportion attributed to their belowground roots and rhizomes [52]. However,

because pelagic *Sargassum* lacks these structures, we have chosen to focus our comparison of macrophytes on rates measured within the phyllosphere (Table 3).

While mass-specific rates of nitrogen fixation within the phyllosphere of macrophytes may not look impressive at first glance, after taking density into account, it is clear diazotrophs in these associations are capable of contributing large quantities of nitrogen to these systems. This is particularly evident for pelagic *Sargassum* which exhibits the highest mass-specific rates (Table 3). Compared to most seagrasses that inhabit shallow coastal areas where terrestrial inputs of reactive nitrogen can be quite high, pelagic *Sargassum* exists in the low-nutrient environment of the open ocean where dissolved inorganic nitrogen is much more limiting. This, along with the lack of rhizomes, means diazotrophic epiphytes within the phyllosphere may be the most important contributor of reactive nitrogen to *Sargassum* itself as well as the community it supports.

## Comparison to planktonic diazotrophs

Planktonic cyanobacterial diazotrophs, including those living symbiotically within diatoms (diatom-diazotroph associations, DDAs), are by far the most widely studied of all marine nitrogen fixers because of their ubiquity throughout the world's oceans. Though the diazotroph community living on pelagic *Sargassum* is epiphytic, the species identified [12, 20, 31, 41] also exist planktonically. Rate comparisons, therefore, can be made as species identity of both habitats overlap.

A comprehensive review of planktonic nitrogen fixation throughout the world's oceans reveals a staggering amount of variability with areal rates ranging from undetectable to over 19,000 µmol N $m^{-2}$ $d^{-1}$ [30]. Recently, Selden et al. [66] measured planktonic rates of up to 42,600 µmol N $m^{-2}$ $d^{-1}$ at the Cape Hatteras front, approximately 125 km northeast of our study site. Even so, epiphytic diazotrophs on pelagic *Sargassum* are among the most productive compared to these depth-integrated rates of nitrogen fixation by planktonic diazotrophs. Though not entirely unexpected, there are several reasons why our rates are on the high end of this range. One may be a consequence of the method used to measure fixation rates. We used the AR technique, which is a proxy measure of *gross* nitrogen fixation, while the more common method in planktonic systems is the $^{15}N_2$ tracer technique, which measures *net* nitrogen fixation, or the amount of fixed nitrogen that has been assimilated. If a certain fraction of this newly fixed nitrogen is released as dissolved organic nitrogen, as is common among diazotrophs [67–69] the $^{15}N_2$ uptake method may underestimate gross fixation rates [70, 71]. For this reason, relative nitrogen fixation rates measured by the AR method are expected to be greater than those measured using $^{15}N_2$ [28]. If the ultimate goal is to understand *total* reactive nitrogen inputs, the AR technique may be more useful as it accounts not just for assimilated nitrogen but also unassimilated nitrogen.

It also seems the relatively high rates of nitrogen fixation by *Sargassum*'s epiphytic diazotrophs may be due to a combination of their marine identity and the sheer density of *Sargassum*, which, like its benthic counterparts, provides extensive surface area for colonization. These qualities directly and indirectly foster high rates of nitrogen fixation, but when combined they make pelagic *Sargassum* a reliable "hotspot" of diazotrophy in the oligotrophic ocean, especially when compared to the relatively dilute planktonic diazotrophs and their ephemeral blooms.

## Considerations and limitations

Over the past several decades, the $^{15}N_2$ tracer method has become the method of choice to measure nitrogen fixation, over the AR technique. Understanding the difference between the

**Table 3. Rates of nitrogen fixation by epiphytic diazotrophs in the phyllosphere of coastal macrophytes.** Rates are highest reported values (unless otherwise noted) under any condition from subtropical to tropical and north to south. Present study is included at the top for reference.

| Location | Macrophyte | Diazotroph(s) identified and/or suggested | N$_2$ fixation | | Method | Comment | Source |
|---|---|---|---|---|---|---|---|
| | | | µmol N g$^{-1}$ h$^{-1}$ min−max* (avg.) | µmol N m$^{-2}$ d$^{-1}$ min−max* (avg.) | | | |
| **Subtropical** | | | | | | | |
| Gulf Stream off Cape Hatteras, North Carolina | Pelagic *Sargassum* | not identified | 0–37.77 (4.156) | 0–30,916 (3,697) | AR, 3:1 | | present study |
| Woods Hole, Massachusetts | *Codium fragile* | *Azotobacter* sp. | 0.52* | – | AR, 3:1 | | [53] |
| Long Island, New York | *Codium fragile* | *Microcoleus lynbyaceus, Calothrix crustacea*, and *Scytonema hoffmannii* | 0.002–0.19 | – | AR, 3:1 | | [54] |
| Beaufort, North Carolina | *Codium decorticatum* | *Calothrix* sp., *Anabaena* sp., *Phormidium* sp. | 0.086* (0.0104) | – | AR, 3:1 | | [55] |
| Catalina Island and Palos Verdes, California | *Macrocystis pyrifera* | heterotrophic bacteria | (0.00738)[a] | – | AR, 3:1 | rates from freshly collected samples | [56] |
| Palos Verdes, California | *Sargassum muticum* | not identified | (0.00625)[a] | – | AR, 3:1 | | |
| Alcudia Bay, Mallorca, Spain | *Posidonia oceanica* | majority of nifH sequences from uncultivated bacteria, some related to *Chlorobium* | (0.149) | 1,090* | AR, 4:1 | | [47] |
| | | diverse assemblage of proteobacteria, firmicutes, and cyanobacteria | – | 3.5–88.7 | AR, 4:1 | | [39] |
| Aqaba, Jordan | *Lobophora* sp. | not identified | – | 82–1,045 | AR, 4:1 | | [57] |
| | *Caulerpa* sp. | | – | 3–445 | | | |
| | *Halophila stipulacea* | phototrophic (winter) and heterotrophic (summer) bacteria | – | 30–6,160 (1,370) | AR, 3:1 | | [38] |
| Homosassa, Florida | *Sargassum filipendula* | *Nostoc* type cyanobacteria | 0–1[b] (0.4) | – | AR, 3:1[c] | | [12] |
| Redfish Bay, Texas | *Thalassia testudinum* | *Calothrix* sp. | 0–11.3 | 0–27,170[d] | AR, 3:1 | | [11] |
| Grand Bahama | *Microdictyon* sp. | filamentous cyanobacteria (e.g., Oscillatoriaceae, *Calothrix*) | 0.22* (0.14) | – | AR, 3:1 | | [58] |
| | | *Calothrix* sp. and non-heterocystous cyanobactera (e.g., *Oscillatoria* sp., *Plectonema*) | (0.0611) | – | AR, 3:1 | | [59] |
| | *Laurencia* sp. | *Calothrix* sp. | (0.564) | – | | | |
| Biscayne Bay, Florida | *Thalassia testudinum* | *Calothrix* sp. | 0.001–0.594 | 357* | AR, 3:1 | | [43] |
| New Providence Island, Bahamas | *Acanthophora* sp. | not identified | (0.2908) | – | AR, 3:1 | dry weights are ash free dry weight | [46] |
| | *Sargassum* sp. 1 | not identified | (0.0784) | – | | | |
| | *Sargassum* sp. 2 | not identified | (0.0472) | – | | | |
| | *Thalassia* sp. | *Calothrix* sp., *Plectonema* sp., *Anabaena* sp. | (0.3478) | – | | | |
| Marathon, Florida | *Syringodium filiforme* | not identified | 0 –trace | – | AR, 3:1[c] | | [60] |
| | *Thalassia testudinum* | | 0–2.57 x 10$^{-5}$ | – | | | |
| Auckland, New Zealand | *Codium* sp. | *Calothrix* sp., *Oscillatoria* sp. | 0.6853* | – | AR, 3:1[c] | | [61] |
| **Tropical** | | | | | | | |
| Praslin, Seychelles | *Sargassum cristaefolium* | not identified | (0.1440) | – | AR, 3:1[c] | | [62] |
| Mahé, Seychelles | *Padina* sp. | not identified | (0.1912) | – | AR, 3:1[c] | | |

(*Continued*)

**Table 3.** (*Continued*)

| Location | Macrophyte | Diazotroph(s) identified and/or suggested | $N_2$ fixation | | Method | Comment | Source |
|---|---|---|---|---|---|---|---|
| | | | µmol N $g^{-1}$ $h^{-1}$ min−max* (avg.) | µmol N $m^{-2}$ $d^{-1}$ min−max* (avg.) | | | |
| Zanzibar, Tanzania | *Sonneratia alba* and *Avicennia marina* | *Rivularia* sp. | – | (790) | AR, 3:1[c] | | [63] |
| | | unidentified (not *Rivularia* sp.) | – | (40) | | | |
| | unidentified seagrasses | *Oscillatoria* sp. and *Microcoleus* sp. | 0.003–0.0183 | | AR, 3:1[c] | | [64] |
| Dar es Salaam, Tanzania | *Halodule uninervis* | filamentous, heterocystous cyanobacteria | 0.01–0.192 | 1.8–18 | AR, 4:1 | | [65] |
| | *Cymodocea rotundata* | | 0.007–0.08 | 1.8–22 | | | |
| | *Thalassodendron ciliatum* | | 0.004–0.061 | 4.1–19 | | | |
| | *Thalassia hemprichii* | | 0.01–0.075 | 4.1–26 | | | |

[a] originally reported as a daily rate, converted to hourly rate (24 h $d^{-1}$)

[b] value taken from figure

[c] $N_2$:$C_2H_4$ ratio not reported, assuming 3:1

[d] originally reported as an hourly rate, converted to daily rate (assuming a photoperiod of 12 h $d^{-1}$)

two is important to the interpretation of this and other studies, especially when comparing rates among studies which employ different methods. Acetylene reduction, a proxy for nitrogen fixation, is a measure of gross nitrogenase activity while the $^{15}N_2$ tracer technique measures net nitrogen fixation, or the amount of fixed nitrogen that has been assimilated [28]. As a proxy, the AR method requires a conversion ratio to translate acetylene reduced to nitrogen fixed. Theoretically, this ratio is 3:1 [72], although inhibition of dihydrogen ($H_2$) production by $C_2H_2$ under experimental conditions enhances the efficiency of the $C_2H_2$ to $C_2H_4$ reduction. A more realistic ratio, accounting for $H_2$ production, is closer to 4:1 [28, 73]. Intriguingly, a wealth of evidence suggests not all fixed nitrogen is assimilated and ratios higher than 4:1 have been used to account for this discrepancy (e.g., [2, 28, 67, 68, 74]). For this study aimed at assessing total reactive nitrogen inputs to the *Sargassum* community, we used the lower 3:1 ratio over those beyond 4:1 as assimilated nitrogen is less important than the total amount of nitrogen entering the environment by this process. As a limiting nutrient, all reactive nitrogen, and especially that which is dissolved, is important to consider. As such, a ratio beyond 4:1, in the context of this study, is unhelpful as it does not consider the nitrogen which is released.

We have taken several precautions to ensure accurate measurements of nitrogen fixation by AR and to reduce experimental artifacts. We limited the assay incubation period to a few hours, added 10–12% (vol/vol) $C_2H_2$ to saturate the headspace, controlled for abiotic $C_2H_4$ production, and ensured equilibration of $C_2H_2$ via agitation [75, 76]. Furthermore, acetylene toxicity has been shown to alter some microbial communities, particularly sulfur and sulfate-reducing bacteria (SRBs; [77]). SRB-dominated diazotroph communities associated with macroalgae are, however, confined to the rhizosphere (e.g., [78, 79]), a structure which pelagic *Sargassum* lacks. For this reason, use of the AR technique is discouraged in benthic systems [77, 80]. In contrast, pelagic *Sargassum*'s microbial diazotroph community is confined to the phyllosphere where cyanobacteria have been found to be the dominant nitrogen fixers (Table 2).

Another important point of consideration for our work is the use of a mat-specific *Sargassum* density to extrapolate our mass-specific rates of nitrogen fixation. This differs from previous extrapolations by Carpenter [20] and Hanson [31], which used Parr's [36] regionally averaged *Sargassum* density values. Although both calculations result in units of fixed N per square meter, they are inherently different rates by virtue of the method used to measure

*Sargassum* density (see "Rate extrapolation" section of Methods). Unlike previous pelagic *Sargassum* work [20, 31], the areal rates presented here are mat-specific and intended to allow comparisons between the rates of nitrogen fixation by the pelagic *Sargassum* community and those of other planktonic and macroalgal systems and to inform biogeochemical models. They do not account for areas of the Atlantic basin where pelagic *Sargassum* is not present, and therefore cannot be extrapolated to the entire Atlantic basin.

## Conclusions

Based on our evaluation of previous pelagic *Sargassum* studies and by comparing our results to those of other epiphytic and planktonic diazotrophs, we posit that nitrogen fixation within the *Sargassum* community is an important source of new marine nitrogen and warrants further study. Given the high mass-specific rates of nitrogen fixation we have observed and the sheer density of *Sargassum* [40], this system likely plays an important role in local, and perhaps regional, nitrogen cycling. Extrapolating our average rate of 4.156 µmol N g$^{-1}$ h$^{-1}$ (an order of magnitude lower than our maximum observed rate), pelagic *Sargassum*'s epiphytic diazotrophs can contribute over 3,500 µmol N m$^{-2}$ d$^{-1}$, a mat-specific rate which rivals that of even the most productive planktonic diazotrophs [2, 30] and far exceeds that of coastal macrophytes (Table 3). Preliminary results of field experiments reveal this estimate may even be on the low end and that *Sargassum*'s diazotrophs likely contribute much more reactive nitrogen *in situ* than laboratory studies can capture (personal observation).

The biogeochemical literature acknowledges a deficit of inputs to the marine nitrogen cycle on the order of teragrams per year [1, 81–83]. Previously overlooked diazotrophs [84] as well as those recently discovered in habitats not formerly considered to host nitrogen fixers [7, 85] have moved the research community past the traditional *Trichodesmium*-centered view, lending credence to the argument that for decades, field studies have completely missed certain populations of diazotrophs and have, consequently, underestimated nitrogen fixation [86].

Given the relative lack of long-term data and rather high rates of nitrogen fixation we have measured, it is possible pelagic *Sargassum*'s epiphytic diazotrophs are among the overlooked and underestimated contributors of reactive nitrogen. The recent expansion and increase of pelagic *Sargassum* and the significant role nitrogen fixation plays in its nutrition and growth [12, 31] further highlight the need for updated knowledge of this system, especially considering the possibility that these epiphytic diazotrophs may, in part, support *Sargassum* blooms [24].

Although frequent sampling over six years within the Gulf Stream off Cape Hatteras, North Carolina has provided a comprehensive view of how rates of nitrogen fixation vary temporally, understanding how these rates differ across *Sargassum*'s wider geographic range remains uncertain. This information is vital to appreciating the true impact pelagic *Sargassum* has on a larger scale. The results of this study, however, suggest *Sargassum*'s epiphytic diazotrophs play a much more significant role in marine nitrogen cycling than previously thought and may supply reactive nitrogen on a scale meaningful to the Atlantic nitrogen budget.

## Supporting information

**S1 File. Mass-specific rates of nitrogen fixation by cruise date.**
(CSV)

## Acknowledgments

We would like to thank the captains and crew of the Albatross Fleet whose local knowledge of the Gulf Stream made this research possible. We also appreciate the invaluable logistical

support of Corey Adams as well as sampling and field documentation by John McCord, Haley Grabner, and Parker Kellam. We thank Stephanie O'Daly, Molly Bost, Maggie Benner, Ted West, Caitlin Seyfried, Mark Stancill, Holly Roberts, Kirsten Morse, Anya Leach, Emma Purinton, and Rebekah Littauer for their assistance in sample collection, processing, and analysis.

## Author Contributions

**Conceptualization:** Claire Johnson, Lindsay L. Dubbs, Michael Piehler.

**Data curation:** Claire Johnson, Michael Piehler.

**Formal analysis:** Claire Johnson, Michael Piehler.

**Funding acquisition:** Claire Johnson, Lindsay L. Dubbs.

**Investigation:** Claire Johnson, Lindsay L. Dubbs.

**Methodology:** Claire Johnson, Lindsay L. Dubbs.

**Project administration:** Claire Johnson, Lindsay L. Dubbs.

**Resources:** Lindsay L. Dubbs.

**Supervision:** Lindsay L. Dubbs.

**Visualization:** Claire Johnson.

**Writing – original draft:** Claire Johnson, Lindsay L. Dubbs.

**Writing – review & editing:** Claire Johnson, Lindsay L. Dubbs, Michael Piehler.

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
