## [Decision Letter · Decision Letter 0]

21 Jun 2023

PONE-D-22-33744Reframing the contribution of pelagic *Sargassum* epiphytic N_2_ fixationPLOS ONE

Dear Dr. Johnson,

Thank you for submitting your manuscript to PLOS ONE. After careful consideration, we feel that it has merit but does not fully meet PLOS ONE’s publication criteria as it currently stands. Therefore, we invite you to submit a revised version of the manuscript that addresses the points raised during the review process.

Based on the reviewers' and my own assessment, I'm thus here inviting you to take all of these comments into careful consideration and to modify your manuscript according to the provided constructive suggestions. I will then be happy to receive and further examine your revised version together with a point-by-point reply to each comment by myself and each reviewer, where you will need to explain any changes done to a particular piece of text, or include supported and convincing counterarguments to any points you may disagree with I'm confident you will find the present comments and suggestions relevant and useful to improve your work and I'm thus looking forward to hearing back form you by the due time.

We look forward to receiving your revised manuscript.

Kind regards,

Marcos Rubal García, PhD

Academic Editor

PLOS ONE

2. We note that Figure 1 in your submission contain [map/satellite] images which may be copyrighted. All PLOS content is published under the Creative Commons Attribution License (CC BY 4.0), which means that the manuscript, images, and Supporting Information files will be freely available online, and any third party is permitted to access, download, copy, distribute, and use these materials in any way, even commercially, with proper attribution. For these reasons, we cannot publish previously copyrighted maps or satellite images created using proprietary data, such as Google software (Google Maps, Street View, and Earth). For more information, see our copyright guidelines: http://journals.plos.org/plosone/s/licenses-and-copyright.

Natural Earth (public domain): http://www.naturalearthdata.com/.

Additional Editor Comments:

Dear authors,

I am very sorry for the long time but, after more than 15 invitations I have only found one referee. based on the report of the referee and my own evaluation of the manuscript I decided that your manuscript is suitable for publication in PLOS ONE after minor changes.

Reviewers' comments:

Reviewer's Responses to Questions

**Comments to the Author**

1. Is the manuscript technically sound, and do the data support the conclusions?

Reviewer #1: Yes

2. Has the statistical analysis been performed appropriately and rigorously? 

Reviewer #1: Yes

3. Have the authors made all data underlying the findings in their manuscript fully available?

Reviewer #1: Yes

4. Is the manuscript presented in an intelligible fashion and written in standard English?

Reviewer #1: Yes

5. Review Comments to the Author

Reviewer #1: This is a significant research manuscript. Research is original and thorough. The writing is clear, and I found virtually no grammatical errors. I suggest a few minor clarifications.

Line 127. In the table, please clarify what n means. I am assuming that it means that either 2 or 3 acetylene reduction samples were run, but clarify in the Table heading.

Line 124. Figure legend 1, what is the black squiggly line? Maybe the edge of the Continental Shelf? If so, state the depth represented by the line. 200 meters?

Line 134 & 148. Can you state the u mol photons per m2/ sec of PAR that the samples were exposed to? This would make it easier for someone to duplicate your research rather than trying to purchase an identical set of light bulbs.

6. PLOS authors have the option to publish the peer review history of their article (what does this mean?). If published, this will include your full peer review and any attached files.

Reviewer #1: **Yes: **Edward J. Carpenter

---

## [Author Response · Author response to Decision Letter 0]

12 Jul 2023

Response to Reviewers (also included in "Response to Reviewers" document)

Dr. García,

Thank you so much for lending your time and expertise to the review of our manuscript and for your persistence in finding a reviewer for our paper—we realize it is not always an easy or straightforward task. We greatly appreciate your service to this journal and your continued efforts toward the publication of our manuscript.

Thank you,

Claire (on behalf of all authors)

Response: I have changed the figure titles and their in-text citations from “Figure” to “Fig” as required.

2. We note that Figure 1 in your submission contain [map/satellite] images which may be copyrighted. All PLOS content is published under the Creative Commons Attribution License (CC BY 4.0), which means that the manuscript, images, and Supporting Information files will be freely available online, and any third party is permitted to access, download, copy, distribute, and use these materials in any way, even commercially, with proper attribution. For these reasons, we cannot publish previously copyrighted maps or satellite images created using proprietary data, such as Google software (Google Maps, Street View, and Earth). For more information, see our copyright guidelines: http://journals.plos.org/plosone/s/licenses-and-copyright.

Response: I have changed the basemap from the ArcMap Light Gray Canvas to USGS’s publicly available “Small-scale Dataset-1:1,000,000-Scale State Boundaries of the United States 201403 Shapefile (available from: https://www.sciencebase.gov/catalog/item/581d052de4b08da350d524e5).

I have also included a citation for the bathymetry contour used in this figure. It was generated from GEBCO’s publicly available 2023 gridded bathymetry data (doi: 10.5285/f98b053b-0cbc-6c23-e053-6c86abc0af7b). I have updated my reference list to reflect this change.

Response: Reference list is complete and correct. I have also added DOIs to all references where they are available.

Reviewers’ comments:

4. Line 127. In the table, please clarify what n means. I am assuming that it means that either 2 or 3 acetylene reduction samples were run, but clarify in the Table heading.

Response: Yes, n stands for the number of replicate assays. This explanation has been added to Table 1’s title.

5. Line 124. Figure legend 1, what is the black squiggly line? Maybe the edge of the Continental Shelf? If so, state the depth represented by the line. 200 meters?

Response: The black like represents the 250-meter isobath. Figure 1’s legend has been updated to include this as well as a citation for the data used to generate it.

6. Line 134 & 148. Can you state the u mol photons per m2/ sec of PAR that the samples were exposed to? This would make it easier for someone to duplicate your research rather than trying to purchase an identical set of light bulbs.

Response: I have replaced the light bulb’s brand name with µmol photons m-2 s-1 PAR in both places.

---

## [Editor Report · Decision Letter 1]

20 Jul 2023

Reframing the contribution of pelagic *Sargassum* epiphytic N_2_ fixation

PONE-D-22-33744R1

Dear Dr. Johnson,

We’re pleased to inform you that your manuscript has been judged scientifically suitable for publication and will be formally accepted for publication once it meets all outstanding technical requirements.

Kind regards,

Marcos Rubal García, PhD

Academic Editor

PLOS ONE
---

## [Editor Report · Acceptance letter]

24 Jul 2023

PONE-D-22-33744R1 

Reframing the contribution of pelagic *Sargassum* epiphytic N_2_ fixation 

Dear Dr. Johnson:

I'm pleased to inform you that your manuscript has been deemed suitable for publication in PLOS ONE. Congratulations! Your manuscript is now with our production department. 

Kind regards, 

on behalf of

Dr. Marcos Rubal García 

Academic Editor

PLOS ONE